# Predictors of Problematic Video Gaming in Elementary School Boys with ADHD

**DOI:** 10.3390/ijerph20136239

**Published:** 2023-06-27

**Authors:** Daniela Cvitković, Karla Stanić, Svjetlana Salkičević Pišonić

**Affiliations:** Faculty of Education and Rehabilitation Sciences, University of Zagreb, Borongajska 83f, 10000 Zagreb, Croatia

**Keywords:** problematic video gaming, ADHD, inattention and hyperactivity/impulsivity, elementary school boys

## Abstract

The aim of the study was to expand the current knowledge base on problematic video gaming and ADHD with four questions: (1) Are there differences in the length of video gaming between boys with ADHD and typically developing boys? (2) Are there differences in problematic video gaming between boys with ADHD and typically developing boys? (3) What are the predictors of problematic video gaming? (4) Does problematic video gaming affect academic performance while controlling for the effect of inattention and hyperactivity/impulsivity symptoms? Symptoms of inattention, hyperactivity/impulsivity scale (SNAP IV), the Problematic Video Game Scale, and the Video Game Patterns Questionnaire were administered to 127 parents of elementary school boys in Croatia, 57 of whom were parents of boys with a clinical ADHD diagnosis and 70 of whom were parents of typically developing boys. The results show that there are no differences in length of play and problematic video gaming between ADHD and typically developing boys. A hierarchical regression analysis showed that age, inattention symptoms, and length of play are predictors of problematic video gaming. Inattention symptoms and age are significant predictors of academic achievement whereas it seems that hyperactivity and impulsivity symptoms are not predictors of problematic video gaming and academic achievement in boys.

## 1. Introduction

### 1.1. Problematic Video Gaming

Gaming is one of the most common forms of entertainment for children. Research in the United States prior to the COVID-19 pandemic shows that elementary school children spend on average up to three hours per day playing video games [1], while Croatian research shows that children spend up to one hour per day [2] or up to two hours per day [3]. At the same time, some children spend three to even six hours a day playing without interruption [2]. In 2021, during COVID-19, video gaming continued to increase dramatically [4].

Studies show that 2–5.5% of adolescents/young adults have a video game addiction [5], and during COVID-19 this number increased. Depending on the country, it ranges from 4.5% in Malaysia to as much as 19% in Italy [6].

The term ‘video game addiction’ or ‘gaming disorder’ and its synonyms such as compulsive, excessive, and problematic video gaming are regularly and interchangeably used [7]. Gaming disorder is defined in the 11th Revision of the International Classification of Diseases (ICD-11) [8] as a pattern of gaming behavior (‘digital-gaming’ or ‘video-gaming’) characterized by impaired control over gaming, increasing priority given to gaming over other activities to the extent that gaming takes precedence over other interests and daily activities, and the continuation or escalation of gaming despite the occurrence of negative consequences. Similarly, Internet gaming disorder (IGD) was included in the Diagnostic and Statistical Manual of Mental Disorders, Fifth Edition (DSM-5), but it was categorized as requiring further research as the disorder is not sufficiently established to be included in the official classification of mental disorders for routine clinical use [9]. Although Internet gaming addiction usually involves the persistent and repetitive playing of video games on the Internet, addiction can also develop in games that do not require the Internet [9]. For this reason, Paulus et.al [10] propose the term ‘computer gaming disorder’ to include playing offline video games on a computer, cell phone, or with consoles because they believe that younger children primarily play games that do not require Internet access.

In this paper we will use the term problematic video gaming.

### 1.2. Impact of Problematic Video Gaming

Gamers with excessive video game use or pathological gamers, children aged 8–18, are likely to have more physical and mental health problems such as hand, neck, and wrist pain [11], attention problems in school [11,12], anxiety, depression, negative self-esteem, psychoactive substance use [13], and involvement in physical fights [11].

Studies emerging from the review by Giardina et al. [14] reported that increases in time spent playing online multiplayer games during lockdown restrictions following the COVID-19 crisis had relaxing effects in the short term, but led to higher stress, anxiety, and depression symptoms in problematic gamers in the long term.

Problematic video game playing is a predictor of behavior problems in both typical children and children diagnosed with ADHD [15].

### 1.3. Problematic Video Gaming in Individuals with ADHD

ADHD is one of the most common neurodevelopmental disorders in children and adolescents [9], characterized by symptoms of inattention, hyperactivity, and impulsivity. Earlier research suggested an association between ADHD symptoms and problematic video gaming. Some studies have examined the correlation between ADHD symptoms and problematic video gaming in children, adolescents, and adults, but without a confirmed diagnosis of ADHD. In the study by Paulus et al. [10], children aged three to seven with higher scores on the inattention and hyperactivity scale also had higher scores on the video addiction scale. Higher scores on the Symptoms of Inattention, Hyperactivity, and Impulsivity Scale are significant predictors of video game addiction in adolescents [16] as well as adults [17]. Several studies have confirmed higher scores on video game addiction questionnaires in children and adolescents diagnosed with ADHD [15,18,19,20,21,22]. In the DSM-5, it is already listed as a comorbid disorder with Internet gaming disorder [9].

Although the correlation between ADHD symptoms and problematic video gaming in children and adolescents has been observed in several studies, it has not yet been clarified. It is unknown to what extent symptoms of inattention and to what extent symptoms of hyperactivity/impulsivity account for more problematic video gaming in children with ADHD. In studies by Masi et al. [15], and Kietglaiwansiri and Chonchaiya [19], both groups of symptoms are significantly correlated with problematic video gaming, but since this is a correlational study, we cannot be sure of the nature of the relationship.

### 1.4. Age, Gender, and Problematic Video Gaming

Research on the relationship between age and problematic video gaming does not provide clear results. Comparisons of the results obtained are difficult because different studies used different measures of problematic video gaming. In addition, some studies have included only online games, while others have included video games that can be played offline. Some studies on a typical population found that video game addiction decreases with age and is lower in adolescents [23,24], one study found no association between age and video game addiction [11], and one found that it increases with age [25]. The relationship between age and problematic video gaming in children with ADHD remains to be researched. The available research confirms clearly that boys are more likely to spend more time playing video games and exhibit more symptoms of problematic video gaming than girls [7,11,13,17,21,25,26].

### 1.5. Patterns of Playing Video Games and Problematic Video Gaming

Several studies have shown that time spent playing video games is a predictor of problematic video gaming, both in children [10] and adult men [17]. A study by Chan and Rabinowitz [27] examined the relationship between symptoms of inattention, hyperactivity, and impulsivity with addiction. Those who play more than one hour per day had more symptoms of inattention and lower grades, while no relationship was found between hyperactivity and playing time. Results on the length of time children and adolescents with ADHD played compared to children with normal development were contradictory. Children with ADHD played video games significantly longer than typical children during both weekdays and weekends [15], whereas in the Bioulac et al. study [18], no difference in game duration or type of preferred games was found between the clinical group of children with ADHD and the control group. In a Thai study [19], there was no difference in play duration on weekends, but children with ADHD played longer during the week.

It has also been noted that among pathological gamers, there are more who have devices available in their bedrooms [11,26]; however, it is not known whether there is an association between device availability and problematic video gaming in children with ADHD.

### 1.6. Academic Achievement and Problematic Video Gaming

Research on the relationship between problematic video gaming and academic achievement does not yield consistent results. Some studies show that video game addiction is a predictor of lower academic achievement [13,16,28], while others have found no association between problematic video gaming and academic achievement [29]. We still do not have a clear confirmation of whether problematic video gaming has a different impact on academic achievement in students with ADHD than in typically developing students. According to the available research, only Haghbin et al. [16] examined whether the association between video addiction and academic achievement differed in terms of symptoms of inattention, hyperactivity, and impulsivity, but the research participants did not have a confirmed diagnosis of ADHD.

### 1.7. Research Aim

The aim of the study was to expand the current knowledge base on problematic video gaming and ADHD. The relationship between ADHD symptoms and problematic video gaming in children and adolescents has not yet been clarified. Moreover, results on the duration of play in children and adolescents with ADHD compared with children with typical development have been inconsistent. It is also not known whether there is an association between device availability and problematic video gaming in children with ADHD. Considering that boys are more likely to play video games and exhibit more symptoms of problematic video game playing than girls, we focus on problematic video gaming in boys in this study.

The first objective was to investigate whether there are differences in the length of video gaming between boys with ADHD and typically developing boys. Based on previous studies, we hypothesized that boys with ADHD spend more hours playing video games.

The second objective is to find out if there is a significant effect of ADHD on problematic video gaming.

The third objective is to investigate possible predictors of problematic video gaming. Several research questions were raised. First, to what extent do age, symptoms of inattention, hyperactivity, and impulsivity contribute to the prediction of problematic video gaming? Second, is there an association between the patterns of playing video games and problematic video gaming, and does an ADHD diagnosis moderate this association?

The fourth objective was to determine whether problematic video gaming has an effect on academic performance, while controlling for the effect of inattention and hyperactivity/impulsivity symptoms.

## 2. Materials and Methods

### 2.1. Sample

Overall, 127 parents of elementary school boys (125 mothers and 2 fathers) from different parts of Croatia participated in the research, 57 of whom were parents of boys with a clinical ADHD diagnosis and 70 of whom were parents of boys with typical development. The age of the boys ranged from 7 to 15; the average age was 10.82 years, 37% lived in rural areas, 31% lived in urban areas with up to 100,000 inhabitants, and 31.5% in urban areas with over 100,000 inhabitants. The sampling method in the research was a non-probabilistic convenience sample.

### 2.2. Data Collection

The questionnaire was sent to the parents of boys with ADHD who used the services of the Teaching and Clinical Center of the Faculty of Education and Rehabilitation at the University of Zagreb. Parents were also approached through Facebook, more specifically through its groups for parents of children with ADHD and groups for parents of children with typical development. The questionnaire was created using Google Forms and respondents could access it through a link during March and April 2022. Answering the questionnaire was possible through a personal computer or a cell phone. This study was approved by the Educational Rehabilitation Studies Committee, University of Zagreb, Faculty of Education and Rehabilitation Sciences. It was conducted in accordance with the Code of Ethics of the Croatian Psychological Society, the Code of Ethics of the University of Zagreb, and Ethics Guidelines for Internet-Mediated Research of the British Psychological Society. Special care was taken to preserve and protect data collected from research participants. At the very beginning of the questionnaire, respondents were informed that all ethical principles would be respected and that their anonymity would be guaranteed.

### 2.3. Measures

To conduct this research, a socio-demographic questionnaire was designed with open- and closed-ended questions that were aligned with the research questions and explored the following: age and gender of the respondent and their child/children, the overall academic achievement of the child in terms of grades, and whether the child has a clinical diagnosis of ADHD and/or other developmental disorders.

#### 2.3.1. Symptoms of Inattention, Hyperactivity/Impulsivity

The Croatian version of the MTA version of the SNAP-IV [30] was used to obtain ratings from parents. The SNAP-IV consists of 26 items that are rated on a 4-point scale (not at all, just a little, quite a bit, very much). The items are divided between three subscales: inattention (nine items), hyperactivity/impulsivity (nine items), and oppositional (eight items). Subscale scores were calculated by creating an average. Higher scores represent more problem symptoms. Items for inattention and hyperactivity/impulsivity can be combined to also create a ‘combined ADHD’ score [31]. Internal consistency appears to be good to excellent. In this study, internal consistency for both the inattention scale and hyperactivity/impulsivity scale is α = 0.95. For this study, inattentive and hyperactivity/impulsivity scales were used.

#### 2.3.2. Patterns of Playing Video Games

The patterns of playing video games were measured using the following: number of electronic devices in the home available to the child for playing video games, available electronic devices in the child’s room, average number of hours played during the week, average number of hours played on weekends, and whether children played more video games during COVID-19.

#### 2.3.3. Problematic Video Gaming

The Croatian adaptation of the Video Game Addiction Test for Parents [32] was used. The original scale has 30 items, including questions about the duration of time playing games during the weekdays and weekends and available devices in the child’s room, which describe patterns of playing video games, and academic achievement. During the translation and standardization, these variables were excluded, as well as seven additional variables from the test. Therefore, the Croatian version of the test had 20 items. Principal component analysis was performed using Oblimin rotation with Kaiser normalization on bigger sample of parents of children with ADHD and typically developed children (n = 186). The factor solution had four main factors extracted, which have accounted for 59% of the total item variance. Despite our data confirming the four factor solution, the cross-loadings of items had made the solution difficult to interpret; therefore, we decided to use the total score, encouraged by very good internal consistency (α = 0.89).

## 3. Statistical Analysis

All the analyses were performed using SPSS 25 for Windows (IBM, Chicago, IL, USA). Descriptive analysis included means and standard deviations. The comparison of the groups was based on one-way ANOVA. The Pearson correlation coefficients determined the bivariate correlation. Hierarchical multiple regression analyses enabled the estimation of the variance, explained by predictor variables on problematic video gaming and academic achievement.

## 4. Results

To meet the objective of the study, a descriptive analysis was first performed. Table 1 shows the descriptive data. The reported average gaming time during the week was slightly less than two hours, while the average gaming time during the weekend was between two and three hours for both groups of respondents. The one-way ANOVA showed that there was no statistically significant difference between the ADHD and control group in the duration of playing video games, both during the week and on weekends, while there was a significant difference in academic achievement.

Table 2 shows the correlation between our predictor and outcome variables. We performed collinearity diagnosis for all the regression analyses, and our tolerance indexes and VIF values were in a range that suggested the collinearity was acceptable (range of tolerance indexes was 0.85–0.99, and range of VIF 1.001–1.175).

We would like to emphasize that there was no significant effect of diagnosis on problematic video gaming in our groups, as can be seen in Table 1.

To investigate the possible predictors of problematic video gaming, a hierarchical regression analysis was performed. In the first step, we entered the child’s age, and, in the second step, symptoms of inattention and hyperactivity/impulsivity (Table 3).

The hierarchical multiple regression revealed that at stage one, age contributed significantly to the regression model, F (1, 125) = 12.66, *p* < 0.001, and accounted for 9.2% of the variation in problematic video gaming. Introducing symptoms of inattention and hyperactivity/impulsivity, the regression model explained an additional 17.7% of the variation, and this change in R^2^ was significant, F (2, 123) = 14.94, *p* < 0.001. When all three independent variables were included in stage three of the regression model, hyperactivity symptoms were not significant predictors of problematic video gaming. The significant predictors were inattention symptoms and age, which suggest that the prevalence of inattention symptoms and higher age are associated with severe problematic video gaming.

To investigate whether patterns of playing video games predict problematic video gaming in boys with ADHD, two hierarchical linear regression analyses were performed for the ADHD and control groups. In the first step, we entered the number of available devices and the presence of devices in the child’s room, and in the second step length of video game playing during the weekdays and weekends (Table 4).

In our ADHD sample, the hierarchical multiple regression revealed that at stage one, the age of boys contributed significantly to the regression model, F (1, 55) = 8.02, *p* < 0.01. Adding the number of video gaming devices and availability of the devices in the child’s bedroom, the regression model explained an additional 10% of the variation in problematic video gaming, and this change in R^2^ was significant, F (3, 53) = 5.20, *p* < 0.05. Adding the length of video gaming during the week and weekend, the regression model explained an additional 389% of the variation in problematic video gaming, and this change in R^2^ was significant, F (5, 51) = 15.61, *p* < 0.001. In our control sample, we found similar results; at stage one, age contributed significantly to the regression model, F (1, 68) = 5.44, *p* < 0.05. Adding the number of video gaming devices and availability of the devices in the child’s bedroom, the regression model explained an additional 15% of the variation in problematic video game playing, and this change in R^2^ was significant, F (3, 66) = 6.43, *p* < 0.01. Adding the length of video gaming during the week and weekend, the regression model explained an additional 41% of the variation in problematic video game playing, and this change in R^2^ was significant, F (5, 64) = 22.16, *p* < 0.001.

In both samples, when all five independent variables were included in stage three, the two variables of the length of play were the only significant predictors of problematic video gaming. The length of play variables seem to behave as suppressor variables in both samples for age, the number of available devices in the ADHD sample, and the presence of a device in the bedroom in the control group. We explain it by the higher correlation coefficients of the length of play variables with the criteria, although the patterns differ with regards to diagnosis. In ADHD children, the number of overall devices seems to be more significant compared to device availability in the bedroom in the control sample, with regards to video game addiction; however, both become suppressed by the length of play. 

To investigate the possible predictors of academic achievement, we performed a hierarchical regression analysis. In the first step, we entered the child’s age, in the second step symptoms of inattention and hyperactivity/impulsivity, and in the third step problematic video gaming (Table 5). 

The hierarchical multiple regression revealed that at stage one, age contributed significantly to the regression model, F (1, 125) = 4.67., *p* < 0.05. Introducing the inattention and hyperactivity/impulsivity symptoms explained an additional 22% of variation, and this change in R^2^ was significant, F (2, 123) = 14.57, *p* < 0.001. Adding problematic video gaming did not contribute significantly to the regression model, F (1, 121) = 0.004, *p* > 0.05. When all four independent variables were included in stage three, the significant, as well as negative, predictors were age and inattention symptoms. The results suggest that the severity of inattention is associated with the problems regarding academic achievement, as it was the stronger predictor, additionally declining with age in our sample. 

## 5. Discussion

The purpose of this study was to compare ADHD-diagnosed boys with typically developing boys, with respect to playing video games, in order to examine possible predictors of problematic video gaming, and to determine whether problematic video gaming has a differential impact on academic performance with respect to an ADHD diagnosis.

There was no difference between the duration of playing during the week and on weekends between the ADHD and control group, which is consistent with Bioulac et al. [18], while in the study by Masi et al. [15], it was found that children with ADHD play video games longer. It is possible that the results were influenced by the research context. Data collection took place during the COVID-19 pandemic. During that time, most education took place online, which meant that children spent more time using computers and cell phones. More time spent at home and the availability of devices led to children playing more video games during the COVID-19 pandemic. In the systematic review of the effects of COVID-19 on child and adolescent gaming addiction [33], it was found that even typically developing children played more video games during the pandemic than before.

In this study, we obtained data on time spent playing video games from parents rather than children, and the results may have been different if we had known the actual time spent playing video games. It is possible that most children, especially children with ADHD, spend more time playing games than parents think, especially if they have the opportunity to play video games in their room.

ADHD diagnosis was not a factor that differentiated our participants in regard to problematic video gaming, which is inconsistent with previous research [15,18,19,21,22]. It is possible that the differences between our results and the results of other studies are due to the fact that other available studies that identified differences in problematic video gaming between the group diagnosed with ADHD and the control group included girls [15,18,19,21,22]; in some studies, there was a disparity in the number of boys and girls between the group diagnosed with ADHD and the control group [15,19,21], and the effect of gender was not controlled for in the clinical sample.

The performed regression has shown that the symptoms of inattention as well as age had significant effects on problematic video gaming. These results suggest that attention difficulties could be a risk factor for problematic video game behavior in the population of boys, which is consistent with a previous study where path analyses showed the direction of causality, and suggest that attention problems may lead to pathological gaming; however, the inverse result was not obtained [34]. Possible reasons for greater problematic video gaming in boys with more symptoms of attention difficulties may lie in the possibility of self-medication through video games. Games might stimulate synaptic dopaminergic transmission similar to methylphenidate stimulation in ADHD, which could lead to improvements in attentional ability [35]. It is also possible that those with greater attention difficulties, in particular, may escape into the world of video games to cope with daily school and study stress, which would be consistent with research findings [35] that problematic video gamers tend to preferentially use dysfunctional coping strategies such as distraction and avoidance.

Our results are in accordance with one previous study by Esposito et al. [25], which found that gaming addiction increases with age. In contrast, studies mostly found a decrease in gaming with age [23,24], or otherwise no relationship between them [11]. Because studies differ in terms of age range, it is difficult to compare the results of measuring problematic video gaming and to draw specific conclusions. 

In addition, we were interested in whether there was a relationship between the pattern of playing video games and problematic video gaming, with regards to ADHD diagnosis. Devices in a child’s room had a moderate effect in the control sample, while the number of available devices had an effect in the ADHD sample in the first step. In both models, the introduction of the length of play variables had a suppressive effect on those variables and thus they emerged as stronger predictors. Our results suggest that for ADHD boys, the length of play during both the week and weekend is associated with problematic video gaming. In the control sample, we found a stronger effect of the length of play during the weekend, when parents are likely to place fewer restrictions, although both are associated with problematic video gaming. Similar results have been found in some studies [11,27]. Even though our final model has not found electronic devices in the child’s room in the control group or the number of available devices in the ADHD group to be significant predictors in our model, we do believe these variables contribute to the video gaming problem and should not be discounted in future research.

Furthermore, we were interested in whether problematic video gaming affects academic achievement. In the complete model, inattention symptoms and age had a statistically significant effect on academic achievement, whereas problematic video gaming and hyperactivity were not significant predictors. Older age is associated with poorer academic achievement which can be explained by the higher demands that students in the higher grades of elementary school are exposed to. Boys with symptoms of attention difficulties, whether or not they have an ADHD diagnosis, have poorer academic achievement, which is to be expected since attention affects all aspects of learning. Similar results were obtained by Haghbin et al. [16]. In Sahin et al.’s study [29], the impact of problematic video gaming on academic achievement was not found, in contrast to some studies that found a negative impact of problematic video gaming on school achievement [13,28]. The studies are difficult to compare because they differ in the way academic achievement is measured. For example, in the study by Brunborg et al. [28], academic achievement was measured by the average of the most recent grades in three subjects. Sahin et al. [29] measured it by the average grades student’s had in the current year. In the study by Van Rooij et al. [13], high school students self-assessed their academic performance.

### Limitations and Future Directions

This is the first study in Croatia to investigate problematic video gaming in children and adolescents diagnosed with ADHD. It is the first study in the available literature in English that has examined the impact of problematic video gaming on academic achievement, as well as the impact of video game device availability on problematic video gaming in individuals diagnosed with ADHD. The strength of this study is that data were collected from different regions of the country, both rural and urban. Another strength of our study design is that data about inattention and hyperactivity, which were collected for the whole sample, have enabled us to investigate these variables in the entire sample, regardless of the diagnosis. 

This study has several limitations. Given the potential for data collection during the COVID-19 pandemic, the survey was conducted on an online sample, and we would suggest replicating it with a larger representative sample. The duration of play was estimated by parents, and it is possible that they did not have insight into all the situations in which their children played video games, which could have influenced the results. On the other hand, there were questions about whether a child’s self-assessment of the duration of video game play is a reliable measure, especially for children with problematic video game behavior and children diagnosed with ADHD, who have difficulty with self-assessment and sense of time. 

The studies are difficult to compare because they differ in terms of the age range of participants included, the measures of problematic video gaming, and academic achievement; therefore, further studies are needed to draw firm conclusions.

## 6. Conclusions

This study shows that attention difficulties, rather than ADHD diagnosis per se, are a significant predictor of problematic video gaming in elementary school boys. The results also show that the duration of play, both during the week and on weekends, is an important predictor of problematic video gaming for both boys with ADHD and typically developing boys. Attention difficulties alone could be a risk factor for poorer academic achievement, regardless of ADHD diagnosis and problematic video gaming. Study findings on problematic video gaming associated with older age suggest a need for the prevention of problematic video gaming at an earlier age. 

## Figures and Tables

**Table 1 ijerph-20-06239-t001:** Comparative data for prediction and outcome variables between ADHD and control group.

	ADHD Group (N = 57)		Control Group (N = 70)		ANOVA
	M	SD	M	SD	F
Child Age	10.70	2.10	10.91	2.17	0.31
Grade Average	4.02	1.01	4.60	0.62	15.90 *
Inattention Symptoms	34.68	4.12	21.79	8.21	116.88 *
Hyperactivity/ Impulsivity Symptoms	32.11	5.77	17.43	6.86	165.658 *
Average Playtime Week	1.75	0.91	1.91	1.07	0.12
Average Playtime Weekend	2.33	1.03	2.67	1.03	1.18
Problematic Video Gaming Scale	50.29	12.91	47.77	12.32	1.27

Note * *p* < 0.001.

**Table 2 ijerph-20-06239-t002:** Correlation between the predictor and outcome variables.

Variables	2	3	4	5
1. Age	−0.19	0.01	−0.17	0.27 *
2. Academic Achievement		−0.43 *	−0.28 *	−0.23 *
3. Attention Symptoms			0.77 *	0.44 *
4. Hyperactivity/Impulsivity Symptoms				0.26 *
5. Problematic Video Gaming				

Note. * *p* < 0.01

**Table 3 ijerph-20-06239-t003:** Results of hierarchical regression analysis predicting problematic video gaming severity from age and symptoms of inattention, hyperactivity, and impulsivity (N = 127).

	Problematic Video Gaming	
Predictors	1st Step	2nd Step
Age	0.30 *	0.27 *
Inattention Symptoms		0.52 **
Hyperactivity/Impulsivity Symptoms		−0.14
R	0.30 *	0.52 **
R^2^	0.09 *	0.27 **
R^2^ Change		0.18

Note. * *p* < 0.01, ** *p* < 0.001.

**Table 4 ijerph-20-06239-t004:** Results of hierarchical regression analysis predicting problematic video gaming from patterns of video game playing in ADHD (N = 57) and control sample (N = 70).

		Problematic Video Gaming				
	ADHD Group			Control Group		
Predictors	1st Step	2nd Step	3rd Step	1st Step	2nd Step	3rd Step
Age	0.36 **	0.34 *	0.13	0.27 *	0.15	−0.06
No. of Available Devices		0.31 *	0.12		0.21	0.10
Presence of Device in Bedroom		−0.15	−0.10		0.29 *	0.05
Length of Play during Week			0.43 **			0.27 *
Length of play during Weekend			0.32 *			0.54 ***
R	0.36 **	0.48 **	0.78 ***	0.27*	0.48 **	0.80 ***
R^2^	0.13 **	0.28 **	0.61 ***	0.07*	0.23 **	0.63 ***
R^2^ change		0.10 *	0.38 ***		0.15 **	0.41 ***

Note. * *p* < 0.05, ** *p* < 0.01, *** *p* < 0.001.

**Table 5 ijerph-20-06239-t005:** Results of hierarchical regression analysis predicting academic achievement (N = 127).

	Academic Achievement		
Predictors	1st Step	2nd Step	3rd Step
Age	−0.19 *	−0.18 *	−0.17 *
Inattention Symptoms		−0.47 ***	−0.47 **
Hyperactivity/Impulsivity Symptoms		0.05	0.05
Problematic video gaming			−.006
R	0.19 *	0.47 ***	0.47 ***
R^2^	0.04	0.22 ***	0.22 ***
R^2^ Change		0.197 ***	0.00

Note. * *p* < 0.05, ** *p* < 0.01, *** *p* < 0.001.

## Data Availability

The data presented in this study are available on request from the corresponding author.

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
