# Peer review of "Predictors of Problematic Video Gaming in Elementary School Boys with ADHD"

_ijerph, 2023, doi:10.3390/ijerph20136239_

Round 1
Reviewer 1 Report
This is a very good paper which is helpful in the context of understanding problematic video gaming in elementary school boys with ADHD. The authors provide an excellent literature review; their design is good; and they address the limitations of the study well in the discussion (such as data collection during the COVID pandemic, which may impact the amount of time that kids were playing video games). This paper is strong, and helps to advance the knowledge in the field. 1) My first comment/question is around ethics approval: the authors document on line 173/174 that "respondents were informed that all ethical principles would be respected and their anonymity would be guaranteed". On lines 400-402, the authors explain that the study received funding from the University of Zagreb, and that there was informed consent. The authors do not address getting ethical approval for doing research with human subjects. 2) There are some small issues with grammar/language - for example: Line 143 - I recommend writing 'typically developing boys' instead of 'normally developing boys'. Also lines 161 and 162, 'lives' should be 'live' (ie 37% live in rural areas) 3) I am confused by the statement on line 311, where the authors write "In our research and systematic review..." and then put reference 33, which has different authors. Either the wording is not right, or the reference is not right. Please clarify.See comments above
Reviewer 2 Report
I was grateful for the opportunity to review this paper - this is an interesting topic, which is likely to be of interest to readers.
I am somewhat concerned about the interpretation of findings, but this may be explained by a typo. Table 3 shows a Negative association between ADHD diagnosis and problematic play in full model , yet this seems to be interpreted as a positive association.
If Table 3 is indeed correct, the suppression effect needs to be explained more thoroughly, and the negative association interpreted. What accounts for the residual variance in ADHD diagnosis once hyperactivity and inattention are taken into account? Do the authors have any measure of family SES (reflecting, perhaps, access to diagnosis)? Additionally, the authors should consider running the model without ADHD diagnosis. The conclusion emphasizes the association between ADHD diagnosis and problematic gaming, but importantly, the ANOVA and step 2 of the regression analysis showed no association between diagnosis and problematic gaming (and the full model an apparently negative association).
Also please consider reporting tolerance and VIF given the correlations between predictors, particularly ADHD diagnosis with hyperactivity and inattention symptoms.
Regarding potential confounders – I do think family SES is important to include if available. Number of devices available, for example, may be confounded with family income, as may access to diagnosis (though admittedly I have no knowledge of the medical system in Croatia).
I would like to see more nuance in the discussion of length of play. It is currently discussed as a risk factor, but is this not a symptom of problematic play? Some more justification for the classification as a risk factor is necessary.
Relatedly, the authors could provide more information on the measure of problematic gaming and its factor structure. Given that the authors found 4 factors, what are the conceptual differences between these factors, and did authors examine them separately at any point? ADHD\inattention may differentially predict different facets of problematic gaming.
In the discussion, the authors highlight differences between their findings and other studies of associations between problematic gaming and academic achievement. How many of the cited studies included innatention symptoms as a control? This is a strength of your study, clarifying that video game play in itself is not associated with academic achievement over and above the effect of inattention.
English language quality is good with only minor editing required. For example, the objectives section contains multiple tense shifts (was to is, etc.)
Reviewer 3 Report
Thank you for the opportunity to review this interesting paper. The research topic is relevant and addresses the current discussion on the role of video gaming for mental health in children.
Two major concerns have come up:
1. The authors seem to assume causality concerning the impact of ADHD on video gaming (e.g., lines 58, 296f, 323f, 371ff, 388), which cannot be met by their research design. In line 89f, they address this problem regarding one study they cited, but I think it applies to the whole paper.
Information on causality cannot be gained from this cross-sectional study. As there is also literature on the effects of video gaming on attention problems, I suggest to refrain from any causal assumptions in terms of “impact”, “risk factors” etc., and refer to more neutral phrases as “correlation”, “association”, etc.. Moreover, I suggest going into more detail about the mutual influence of attention problems and video gaming (e.g., children with inattention symptoms may feel more attracted to video gaming, or parents of ADHD children may allow their kids to play more video games to calm them down).
2. Research Aim and Results: Some of the constructs the authors presented are highly intercorrelated (e.g., ADHD diagnosis and inattention symptoms; patterns of video game playing and video gaming addiction), and I do not feel comfortable with some of the regression analyses treating them as independent predictors (table 3, 5) or as predictors and outcomes (table 4). What the authors refer to as suppression effects seems to be a problem of multicollinearity, even if the multicollinearity statistics were acceptable. I suggest omitting ADHD diagnosis from the regression analyses in table 3 and 5.
Minor comments:
l 123f: “some observations of typically developing students show that video game addiction is a predictor ….” – in the presence of video game addiction, the term “typically developing students” (meaning students without the presence of mental health problems?) seems inappropriate
l83, l133: once the abbreviation “ADHD” has been introduced, the term “attention deficit hyperactivity disorder” no longer has to be written out in its entirety
l146ff, l152ff: the second and third research questions relate to … “boys with ADHD” (l147/l153), which implies that analyses will be only carried out with the subgroup of boys with ADHD. As ADHD diagnosis is introduced as a moderator in both research questions, the phrasing seems contradictory. I suggest omitting “in boys with ADHD” in line 147 and line 153
l198 ff: It would be helpful to see some item examples and/or the interpretations of the four main factors that were extracted in order to better understand the contents of this questionnaire.
l296f: “The results suggest …” should not be part of the results section, but of the discussion. (and it reflects the causal assumptions that cannot be met by your research design)
l382: ADHD disorder: eliminate “disorder” as it is already part of the term “ADHD”
I suggest proof-reading by an English native speaker; as I am no native speaker myself, I refrain from further suggestions concerning phrasing and grammar.
Round 2
Reviewer 2 Report
I thank the authors for their attention to my comments, in particular my main comment about the interpretation of the association with ADHD diagnosis.
You have added a t-test, though I do not think this is necessary given your ANOVA results (and it becomes confusing as you report different means and SDs for the gaming scale in the ANOVA and t-test - please verify data and choose one test to present).
The labeling in the tables should be consistent - I assume "video game addiction" in table 4 refers to the problematic gaming scale?
I would also consider adding age as a covariate in the models presented table 4.
I thank the authors for explaining the (lack of) associations with measures of SES, however I think it would be good practice to also include these as covariates in the regression models, or show associations in the correlation tables, if just to humour me .
The authors have mostly done well to avoid causal language, though there are still some instances where causality is implied (e.g., in the last paragraph of the results, where the authors state that inattention is the "underlying problem in academic achievement").
I suggest a quick english language edit for flow and clarity.
Reviewer 3 Report
Thank you for implementing my suggestions.
Author Response
Thank you for your suggestions.